# Advancing clinical cohort selection with genomics analysis on a distributed platform

Jaclyn M. Smith[1,2]*, Melvin Lathara[2], Hollis Wright[2], Brian Hill[2,3], Nalini Ganapati[2], Ganapati Srinivasa[2], Christopher T. Denny[4,5,6,7]

1 Department of Computer Science, University of Oxford, Oxford, United Kingdom, 2 Omics Data Automation Inc., Beaverton, Oregon, United States of America, 3 Department of Computer Science, University of California, Los Angeles, California, United States of America, 4 Division of Hematology/ Oncology, Department of Pediatrics, Gwynne Hazen Cherry Memorial Laboratories, University of California, Los Angeles, California, United States of America, 5 Molecular Biology Institute, University of California, Los Angeles, California, United States of America, 6 Jonsson Comprehensive Cancer Center, University of California, Los Angeles, California, United States of America, 7 California NanoSystems Institute, University of California, Los Angeles, California, United States of America

* jaclyn.smith@cs.ox.ac.uk

**Data Availability Statement:** The minimal dataset consisting of the raw runtimes from the figures have been made available at 10.6084/m9.figshare. 11796126. The Phase III 1000 genomes whole exome sequencing data are available freely for

## Abstract

The affordability of next-generation genomic sequencing and the improvement of medical data management have contributed largely to the evolution of biological analysis from both a clinical and research perspective. Precision medicine is a response to these advancements that places individuals into better-defined subsets based on shared clinical and genetic features. The identification of personalized diagnosis and treatment options is dependent on the ability to draw insights from large-scale, multi-modal analysis of biomedical datasets. Driven by a real use case, we premise that platforms that support precision medicine analysis should maintain data in their optimal data stores, should support distributed storage and query mechanisms, and should scale as more samples are added to the system. We extended a genomics-based columnar data store, GenomicsDB, for ease of use within a distributed analytics platform for clinical and genomic data integration, known as the ODA framework. The framework supports interaction from an i2b2 plugin as well as a notebook environment. We show that the ODA framework exhibits worst-case linear scaling for array size (storage), import time (data construction), and query time for an increasing number of samples. We go on to show worst-case linear time for both import of clinical data and aggregate query execution time within a distributed environment. This work highlights the integration of a distributed genomic database with a distributed compute environment to support scalable and efficient precision medicine queries from a HIPAA-compliant, cohort system in a real-world setting. The ODA framework is currently deployed in production to support precision medicine exploration and analysis from clinicians and researchers at UCLA David Geffen School of Medicine.

## Introduction

The affordability of next-generation genomic sequencing and the improvement of medical data management have contributed largely to the evolution of biological analysis from both a

download from the International Genome Sample Resource (https://www.internationalgenome.org/data-portal/data-collection/phase-3). At this time, AtLAS data cannot be shared publicly in an unrestricted fashion due to institutional restrictions related to UCLA's investigations review board (IRB). For further information, follow up inquiries can be made to CLajonchere@mednet.ucla.edu. We have made a version of the i2b2 plugin available under the MIT license, and the GDBSparkAPI and gdb-mapping database available for academic use. These components can be found at https://github.com/OmicsDataAutomation/i2b2-oda-framework. All GenomicsDB components are available at: https://github.com/GenomicsDB/GenomicsDB for free academic use under the MIT license. I2B2 patient querying related components are available at https://www.I2B2.org/software/index.html.

**Funding:** C Denny is supported by the National Center for Advancing Translational Sciences, National Institutes of Health, through the University of California, Los Angeles, Clinical and Translational Science Institute, under award number UL1TR000124 and UL1TR001881. All other support was provided through Omics Data Automation, Inc. Omics Data Automation employees (J Smith, M Lathara, H Wright, B Hill, N Ganapati, and G Srinivasa) were supported by the National Science Foundation under award number 1721343 and via work contract from UCLA David Geffen School of Medicine. Detailed contributions from ODA employees are described in the author's contributions section.

**Competing interests:** J Smith, M Lathara, H Wright, B Hill, N Ganapati, and G Srinivasa are sponsored by Omics Data Automation, Inc. Competing interests have been fully disclosed and arranged through contract and licensing agreements with UCLA David Geffen School of Medicine. This does not alter our adherence to PLOS ONE policies on sharing data and materials. The authors declare that they have no other competing interests.

clinical and research perspective. Precision medicine is a response to these advancements that aims to tailor a medical treatment to an individual based on their genetic, lifestyle, and environmental risk factors [1]. While current medical practice is limited to using broad populations with heterogeneous characteristics, precision medicine places individuals into better-defined subsets based on shared clinical and genetic features. This fine-tuned, cohort-based method determines relative risk factors and potential therapeutic responses with higher accuracy [2]. Though a promising field, the identification of personalized diagnosis and treatment options is dependent on the ability to draw insights from large-scale, multi-modal analysis of biomedical datasets.

The integration of high-throughput genomic sequencing data and electronic health record (EHR) derived, phenotypic data is at the core of precision medicine efforts. Even before integration, genomic and clinical data each have specific bottlenecks that impede effective utilization of these data in practical analysis. EHR data requires extensive cleaning and restructuring for use in cohort analysis and clinical trial identification. This can be accomplished through ETL (extract transform load) and indexing procedures to process the data into a form that can be efficiently queried from a relational database [3]. Informatics for Integrating Biology and Beside (i2b2) is a framework that enables cohort exploration and selection on clinical attributes, such as International Classification of Disease 10$^{th}$ revision (ICD10) codes [4,5]. The i2b2 framework is made up of series of components that work together to query and analyze clinical data. One such cell is the clinical research chart (CRC) that queries a relational database that stores ontological and clinical data with a patient-centric, star schema [4,5]. This system has been widely deployed for clinical data exploration in hospitals across the United States [6], and supports drag-and-drop, clinical cohort queries that interact with the backend relational database through a browser-based user interface.

Genomic data are large, heavily sparse, and in general, inefficiently stored in relational format. Columnar data stores can be specialized for sparse, multidimensional array representation to provide a scalable means to load, store, and query genomic variant data [7,8]. One such columnar data store is GenomicsDB [9], which has exhibited linear import and query execution time with respect to sample size [7]. The power of GenomicsDB has warranted the use of the database in the Genomics Analysis Toolkit (GATK) since version 4.0 [10,11] as a more efficient alternative to flat files. Variant data can be visualized in GenomicsDB as a sparse, two-dimensional matrix with genomic positions on the horizontal axis and samples on the vertical axis (Fig 1). Under this representation, columns can maintain top-level information about the variant, such as genomic position and reference allele. Cells of the matrix store data about the sample for the given position, such as genotype call, read depth, and quality scores. A single matrix instance has several vertically partitioned segments, called arrays, which support contiguous storage of genomic regions on disk. These arrays can be split into several partitions, thus providing support for data distribution [7].

To understand how clinical and genomic data can be combined for use in cohort selection, consider a use case from UCLA David Geffen School of Medicine. Prior to the efforts discussed later in this paper, UCLA had an existing i2b2 system deployed for the purposes of performing clinical queries and retrieving a patient set count. Requirements for this system include a rolling update of clinical data, the processing and storage of associated genomic sequencing data, and patient consent to share such data. The ultimate goal was to query the genomic data from within the i2b2 interface along side the clinical attributes. The genomic and clinical data would be accessible to both clinicians and researchers for de-identified, cohort exploration and selection. At the outset of the efforts discussed in this paper, genomic data were stored as individual flat files that were siloed from the i2b2 system. UCLA

**Fig 1. Conceptual view of GenomicsDB.** Variant data is represented as a sparse two-dimensional matrix, and can be split into multiple, vertically partitioned arrays for distribution.

projected an accrual of up to 1000 new samples per week that would need to be imported into and accessible from this system.

Given the systematic use case described above, we can now consider the types of queries that would be proposed to such a system. A user may be interested in exploring genotype information for patients who have been diagnosed with a mental health disorder. The user would like to look at reference and alternate allele counts for two genes that have previously shown association to a mental health disorder (Fig 2). Alternatively, a user may be interested in creating clinical subgroups based on various medications prescribed at the hospital. This user is further interested in patients diagnosed with malignant breast neoplasms, and requests to zone in on two point mutations previously known to associate with breast cancer (Fig 3). Such queries can be submitted to the system as an exploratory tool for clinicians to dive deeper into genotype-phenotype associations, explore targeted-treatment options, or estimate sample size for a proposed clinical trial. These queries can enable hypothesis exploration for researchers, who will then switch to an advanced interface for detailed analysis.

I2b2 has a modular design that makes the framework easy to extend with backend features. Plugins have been developed previously that enable querying of genomic data from within i2b2 for datasets with hundreds of samples [6,12]. Given the projected accrual of up to 1000 new samples per week noted above, a scalable genomics database as well as an efficient processing environment was required to manage and analyze the data at UCLA. Since clinical data and genomic data are optimally stored in databases natural for their specific data structures, a scalable solution to clinical-genomic data integration should leave the data in the respective optimal data store and provide the mechanisms to perform efficient aggregate queries from these sources. Given this, we built a system that would (i) integrate genomic sequencing data with the existing i2b2 instance at UCLA, (ii) maintain the data sources in their respective data stores, and (iii) support efficient and scalable integrative analysis by means of a distributed processing environment.

Distributed processing platforms, such as Apache Spark [13,14], are becoming increasingly popular for large-scale, genomic data analysis [15,16,17]. Spark provides a powerful, programmatic interface that abstracts the distribution from the user. A Spark cluster consists of a

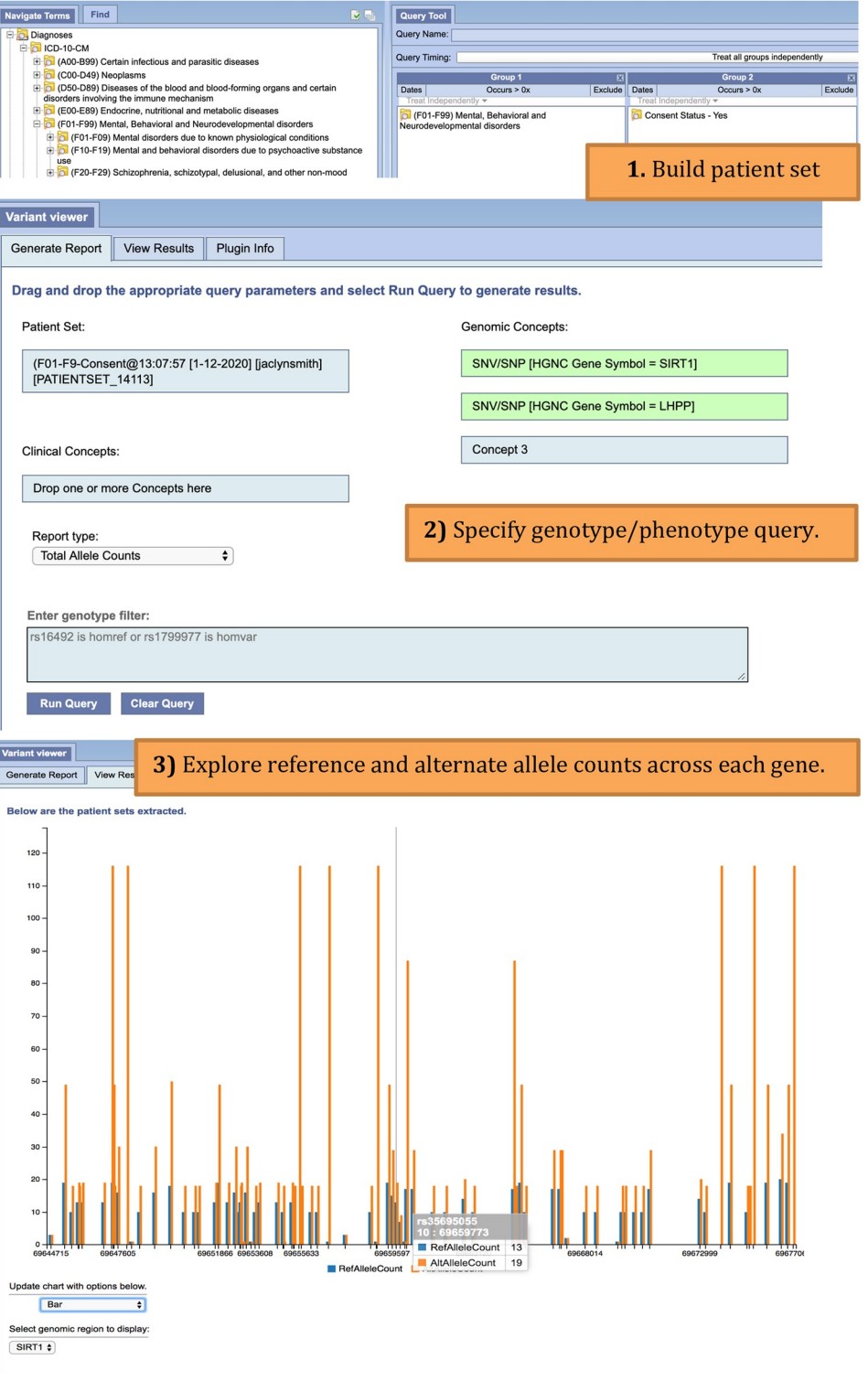

**Fig 2. Allele counts i2b2 workflow.** User starts by building a patient set (1) that includes patients diagnosed with mental, behavioral, and neurodevelopmental disorders (ICD-10-CM, F01-F99). The user also selects to return patients who have consented to share genotype information. The query goes on to request total allele counts across SIRT1 and LHPP genes (2). The total allele counts are returned in report format for each gene (3). The user can switch between gene views and change the type of plot.

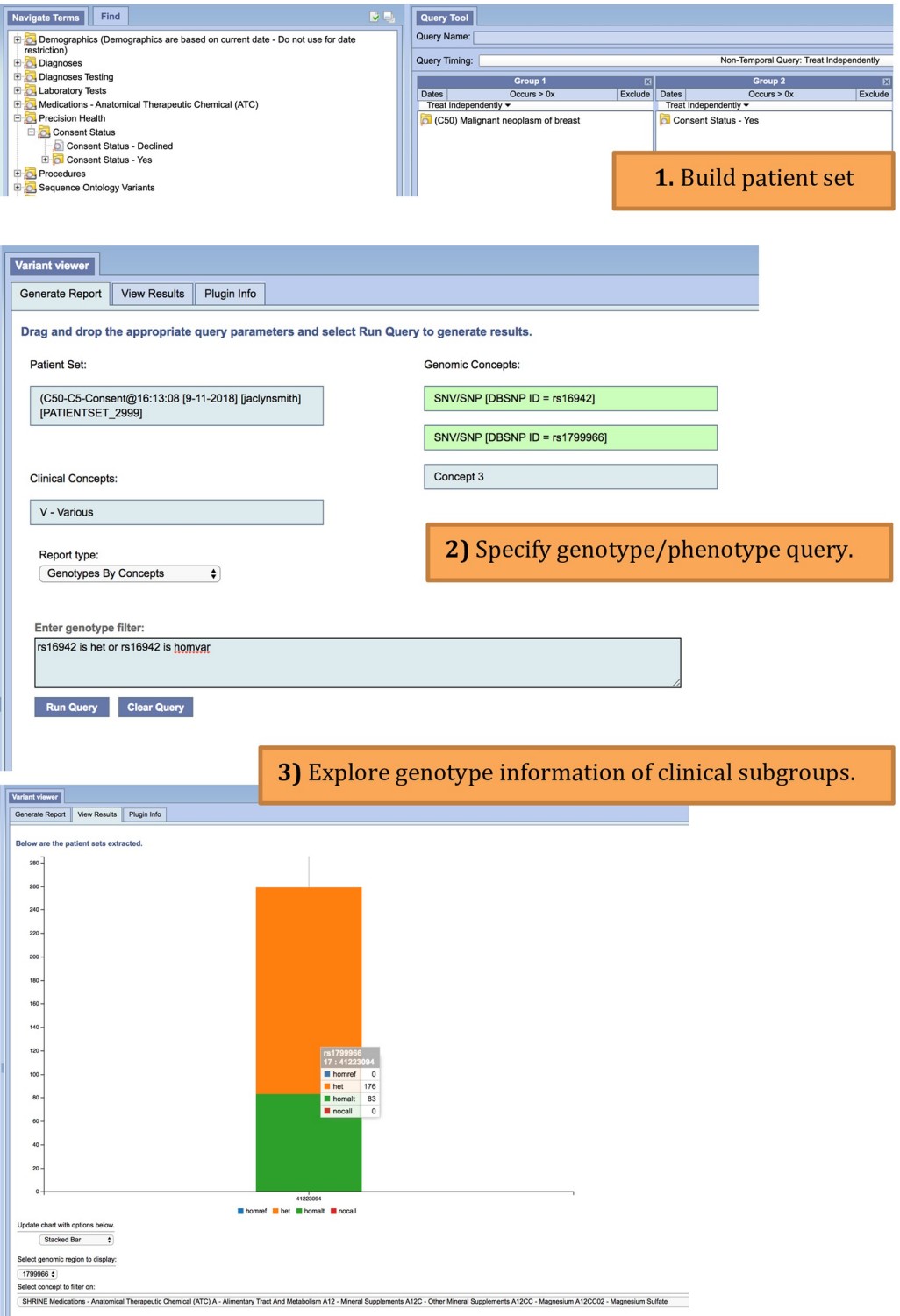

**Fig 3. Genotype by concept i2b2 workflow.** User starts by building a patient set (1) that includes patients diagnosed with malignant neoplasm of breast (ICD-10-CM, C50). The user also selects to return patients who have consented to share genotype information. The query goes on to request genotype distributions for two mutations (rs16942, rs1799966). The query specifies to group by various medications prescribed (2). The genotype distributions are returned in report format for each genomic position, and for each clinical subgroup (3). The user can switch between genomic positions, clinical subgroups, and change the type of plot.

master node and a set of worker (slave) nodes. The master node delegates tasks to the worker nodes, which will execute relevant tasks over the distributed datasets. In the event that a worker fails, Spark uses a data structure known as Resilient Distributed Datasets (RDDs), which will reallocate data to other nodes and ensure nothing is lost (fault-tolerance) [18]. Distributed file systems, such as the Hadoop Distributed File System (HDFS) [19] and Amazon Elastic MapReduce File System (EMRFS) [20], are often used in conjunction with such platforms to maintain data integrity and high data throughput. In contrast to distributed file systems, a local file system does not allow worker nodes to have centralized access to all the data required for the application. Similarly, network file systems provide limitations since all data is physically stored on a single machine and not distributed.

Prior to the work described in this paper, GenomicsDB supported querying of genomic data from Spark [7], but did not support the reading and writing of genomic data from genomic arrays stored on a distributed file system (legacy mode). Without support for a distributed file system, the process of querying GenomicsDB from Spark meant that arrays had to be manual organized across the worker nodes. The worker could only access the genomic data that existed physically on that node. Under the organization of legacy mode, a query would be broadcast to each worker node, executed, and then a single RDD partition would be loaded with all the genomic variant data available to that worker.

There are several restrictions with this configuration that impact both genomic data storage and the Spark distribution abilities. First, a worker node is required to store the genomic data locally meaning the node needs enough space to store the genomic data and write out temporary work files from Spark. The addition of more genomic data to the system could require a resizing of the worker nodes and lead to system downtime. Second, a worker node can only query one GenomicsDB array and can only load data from this array into a single RDD partition. This limits the distribution power of Spark since the number of RDD partitions should at least equal the number of cores available to an application to take full advantage of the available resources. Finally, if a worker node fails in legacy mode, the data must be reloaded or copied back to the node from an archive. This reduces the fault tolerance power of Spark and makes auto scaling the cluster a difficult task. Auto scaling is the ability to increase or decrease the number of worker nodes according to application load and is an important feature for resource management in Spark. Fortunately, the issues associated with Spark and legacy mode of GenomicsDB can be addressed by extending GenomicsDB to work with a distributed file system.

In response to the issues described above, we have extended GenomicsDB to support reading and writing to/from a distributed file system, such as HDFS and Amazon Simple Storage Service (Amazon S3) [21]. This setup better utilizes the distributed power of both Apache Spark and GenomicsDB, reduces the space requirements for a worker node, and maintains the fault-tolerant behavior of an RDD. These extensions, together termed the Omics Data Automation (ODA) framework, have been used to create a precision medicine platform that enables integration and distributed aggregation of EHR-based clinical data and associated genetic data.

In this paper, we show that the ODA framework enables GenomicsDB to exhibit worst-case linear scaling for array size (storage), import time (data construction), and query time. We go on to show that the ODA Framework also exhibits worst-case linear time for both import of clinical data and aggregate query execution time within a distributed environment. This work highlights the integration of a distributed genomic database with a distributed compute environment to support scalable and efficient precision medicine queries from a HIPAA-compliant (Health Insurance Portability and Accountability Act of 1996), cohort system in a real-world setting. The ODA framework is currently deployed in production for use by both

clinicians and researchers at UCLA David Geffen School of Medicine and the extended version of GenomicsDB is openly available at www.genomicsdb.org.

## Materials and methods

The ODA framework is responsible for genomic variant data storage in GenomicsDB, maintaining mapping information to a clinical data store, and enabling users to perform Spark-based queries to the platform from a graphical interface or a programmatic interface. Queries executed on the framework (i) consolidate the requested clinical and genomic data (GenomicsDB) in a distributed environment (Apache Spark), (ii) perform the aggregate calculation within the distributed environment, and (iii) return the results to the user. Based on the UCLA use case, these components were integrated with an i2b2-based, cohort system in a HIPAA-compliant, protected subnet. The UCLA implementation is deployed on Amazon Web Services (AWS) [22], which leverage AWS provided subnets and Amazon's Elastic MapReduce (EMR) instances [20]. Fig 4 presents a synergistic view of the ODA framework and i2b2 within a protected subnet. Focus on AWS services is attributed to the UCLA implementation, and it should be noted that the ODA framework is designed to be agnostic to any cloud provider or local hardware.

### Data storage

GenomicsDB was augmented to enable writing and reading of arrays on HDFS-compliant file systems, in addition to existing POSIX (Portable Operating System Interface) support. Genomic variant data read from Variant Call Format (VCF) [23] files are imported into GenomicsDB arrays residing on a distributed file system through the standard GenomicsDB VCF import process. The import process uses a set of configuration files and Samtools HTSLib (High-throughput sequencing library) [24] to read block-compressed and indexed VCF files. This process writes to several GenomicsDB arrays at once with the use of GNU parallel [25], which maintains an independent import process for each array. Performance for parallel import of GenomicsDB arrays is dependent on 1) the number of arrays produced, and 2) the resources available to the ETL process, such as the number of cores, memory and network throughput. Once the arrays are loaded, the VCF files are no longer needed and can be moved to a cold storage archive. For the UCLA import process, the genome was split into 1000 evenly sized sections to produce 1000 GenomicsDB arrays. These arrays are stored on Amazon S3 and are accessed from an Amazon EMR instance via AWS EMRFS distributed file system. The HDFS-compliant additions to GenomicsDB have been integrated into the GenomicsDB github repository.

Clinical data is maintained in a database and application chosen, or already in-use, by the medical institution. To ensure HIPAA-compliance, the ODA framework acquires clinical data by enforcing encryption at REST (Representational State Transfer) with SSL (Secure Sockets Layer) for all network-based database communication. Mapping information is used to associate clinical data to samples stored in GenomicsDB. The mapping information constructs a relationship between a de-identified patient id and the relative genomic sample information. The patient identifier is sourced from the clinical data store, such as i2b2. This information, along with the configuration files used in the GenomicsDB ETL process, is used to load a mapping database. This mapping database also maintains metadata on GenomicsDB arrays. In essence, the mapping database provides a global view over the relevant data sources required to perform an integrative query over the clinical and genomic information. The mapping database (PostgreSQL [26]) comes with a core relational schema and a python interface for ease of database construction and maintenance. The UCLA implementation uses i2b2 with a

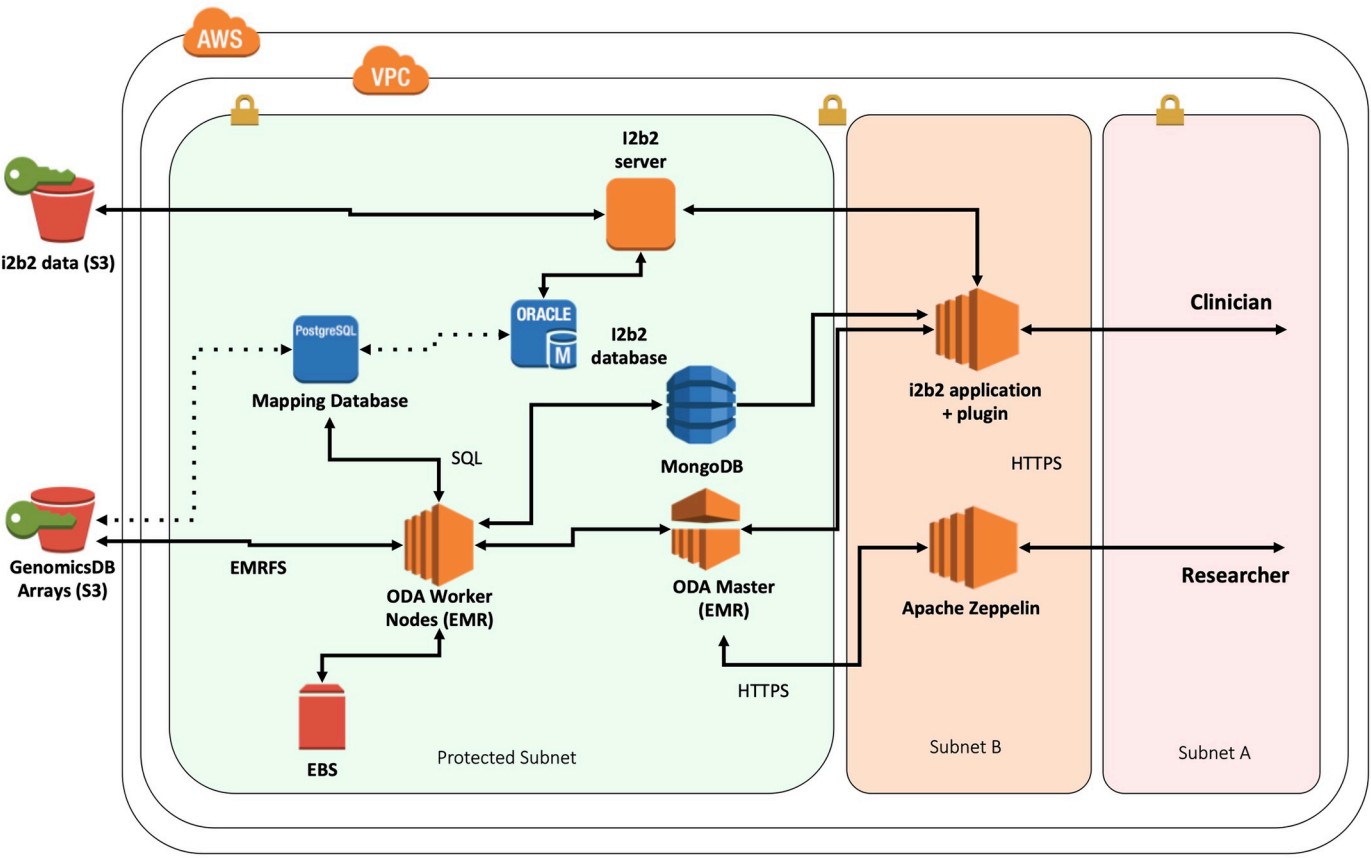

**Fig 4. ODA framework diagram.** System is deployed in a HIPAA-compliant AWS environment, including the connections with the clinical data store (i2b2). The dotted lines are conceptual links between the relative data stores.

PostgreSQL backend as their clinical data store and stores mapping information in a separate PostgreSQL mapping database.

## Apache spark

To support distributed queries and processing, the majority of the query and analytical components reside on the master of the Spark cluster. For the UCLA implementation, the Spark cluster is an AWS Elastic Map Reduce (EMR) instance. The EMR instance contains a Spark master that delegates tasks to worker nodes. The master node serves as the entry point to queries that are sent out to the worker nodes and in the process, loads data from each of the GenomicsDB arrays into Spark RDD partitions. Since GenomicsDB arrays are made available to the worker via a direct connection to a distributed file system, any of the GenomicsDB arrays are accessible to all worker nodes in the cluster. The master node is responsible for delegating query tasks to the workers and these query tasks are responsible for loading of genomic variant data into Spark RDDs.

When a user submits a query to the ODA framework, the Spark master decomposes the query into a list of smaller queries that are distributed to the worker nodes as query tasks. The worker nodes perform the query tasks assigned to them, which is some subset of the whole query list. The number and size of query tasks is proportional to the number of GenomicsDB arrays, such that each query task will query one GenomicsDB array and load the result into an

RDD partition. The collection of RDD partitions across all workers collectively contains all the genomic information queried from GenomicsDB. This means the number of GenomicsDB arrays produced during the import process also helps balance the query workload. The optimal distribution of genomic variant is determined by balance of file open and read operations as well as RDD partition size. More distribution will lead to an increase in open and read operations and smaller RDD partitions, where as less distribution can cause more overhead for smaller query regions by creating large RDD partitions.

## Programmatic access

The GDBSpark API is an integral component of the ODA framework that provides support for distributed querying, loading, and aggregating clinical, genomic, and relevant mapping data sources. Clinical data accessed through the API will require a data handler to tell the API how to interact with the source data. For the UCLA implementation, the API contains an i2b2 data handler that supports the import of an i2b2 XML (eXtensible Markup Language) file received from the CRC cell. The GDBSpark API is implemented in Scala 2.11. This API represents genomic variant data as a VariantContext object provided from HTSLib [24] as specified in the GenomicsDB Java Native Interface (JNI). The API acts as an intermediate layer between Spark, GenomicsDB, and the mapping database to query genomic data, load Spark RDDs, and associate to the clinical dataset for downstream computation. In general, the clinical data from the CRC cell of i2b2 and the aggregation of the clinical data with genomic information in VariantContext RDDs are distributed across the worker nodes. The API also provides a genomic toolkit to perform pre-defined aggregate statistics for genomic and clinical data.

## User interfaces

There are two analytical interface components that interact with the ODA framework to support distributed analytics, an Apache Zeppelin interactive notebook environment [27] and an i2b2 plugin, VariantView. The VariantView plugin (Figs 2 and 3) is designed to perform aggregate clinical and genomic queries from within the i2b2 interface. A user first creates an i2b2 patient set from the "Find Patients" window. This patient set is then referenced in the VariantView plugin Generate Report Tab along with additional clinical attributes (i.e. ICD10 code), and genomic regions (dbSNP rs-identifiers and attributes from the Sequence Ontology). Users have the ability to filter by genotype specified in the form: rs### (is | is not) (homref | homvar | het | nocall). The aggregate queries are predefined as *reports*, which specify the aggregate calculation to perform when the Spark application is submitted. There are three kinds of reports are available in the UCLA instance: total allele counts (Fig 2), genotype distribution by clinical attribute (Fig 3), and genomic-only based cohorts. The results from these reports are displayed in the View Results tab.

The VariantView i2b2 plugin consists of two main components: a frontend, graphical user interface that exists within the i2b2 browser, and a backend that extends the i2b2 hive with an additional service (cell). The frontend uses the i2b2 JavaScript API to communicate with the backend through plugin-specific, XML messages supported by the i2b2 REST API. The backend of the VariantView plugin is responsible for submitting a Spark application to the ODA framework with SparkLauncher. SparkLauncher enables i2b2 to run spark-submit, sending the application jar and associated parameters to the AWS EMR instance that houses the ODA framework. The incoming job requests from i2b2 are scheduled with Hadoop YARN [19]. Once the application finishes aggregating the results of both queries, the results are written back to MongoDB [28] using SSL with a unique identifier. The backend of VariantView is notified of the success or failure of the spark job. On success, the plugin backend will query

MongoDB for the results using the unique identifier. The results are then formatted into a plugin-specific, XML response message and sent to the frontend. The frontend accesses the required information from the response and plots the results using a JavaScript plotting library (d3.js) [29]. If the spark application reports failure, the response will contain error information that will then be displayed to the user. The backend of VariantView is written in Scala 2.11 and is based on a tutorial plugin provided by the i2b2 [30].

Fig 5 shows the flow of a query originating from the VariantView plugin. This entry point could be any Spark-based application that submits to the ODA framework with the GDBSpark API, such as the Zeppelin notebook environment. This notebook environment was added to the framework to enable users to develop code and execute distributed queries to GenomicsDB without having to be concerned about the configuration of a Spark cluster. The Zeppelin instance at UCLA comes with several notebooks that support use cases proposed from clinical geneticists, clinicians, and bioinformaticians at David Geffen School of Medicine. Users are able to create their own notebooks and develop their own analysis within this environment. The analyses and potential additional system requirements for optimizing these workloads are beyond the scope of this work, so we omit further discussion about the Zeppelin interface in the rest of the paper.

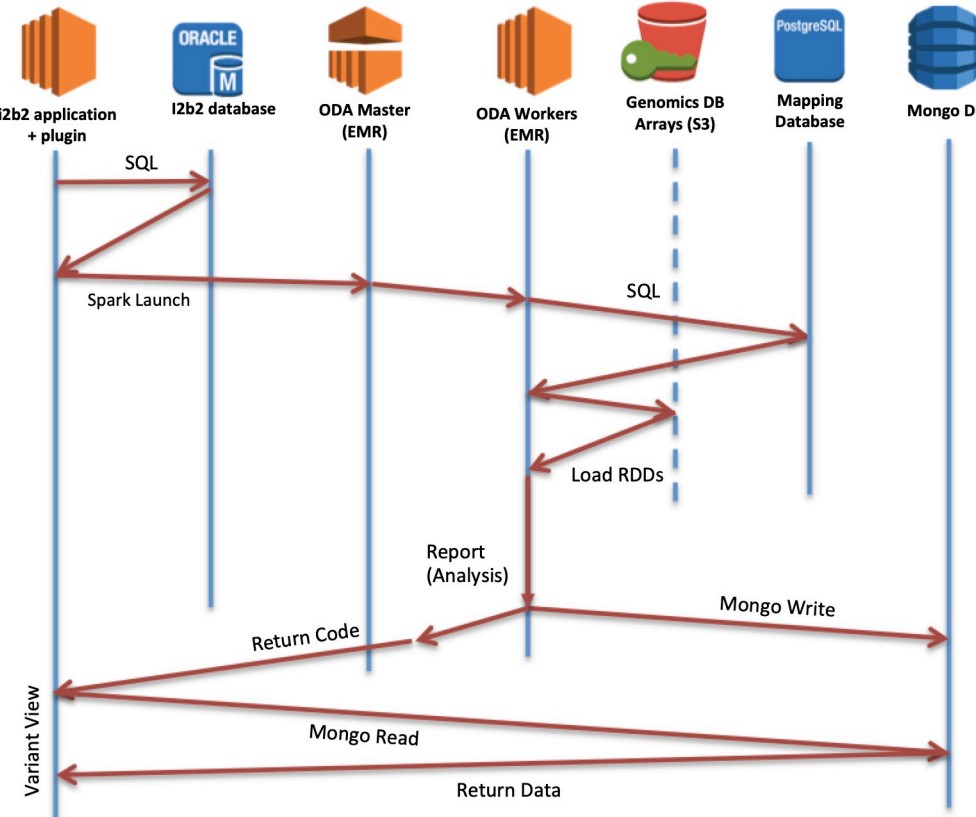

**Fig 5. Query timeline.** Query starts when a user makes a query form the VariantView i2b2 frontend for clinical attributes. Plugin submits a Spark job with the clinical info and requested analysis from the user. Master delegates to workers, which collate mapping information, query GenomicsDB, and load variants into RDDs. User-specified report is performed in Spark. Results are written to MongoDB and a return code is simultaneously sent to the VariantView plugin and visualized in i2b2 browser.

## Results

The following describes the results of a set of experiments designed to highlight efficient variant data loading, storage, and querying of genomic data, stored in GenomicsDB, to/from the distributed platform for an increasing number of samples. The timing to import clinical data as well as the time to aggregate the clinical and genomic data is also reported. All experiments were performed in AWS EMR- 5.7.0, with Spark 2.1.1, and Scala 2.11. Two datasets are referenced throughout the experiments: the Phase III 1000 Genomes Whole Exome Sequencing (WES) dataset [31] and the AtLAs target sequencing data [32]. The AtLAs dataset was generated using a microarray technology to interrogate approximately 600,000 genomic loci. The AtLAs dataset is the UCLA patient data that has associated clinical data stored in i2b2. By contrast, the 1000 genomes data was generated using a high-throughput, whole exome sequencing platform that targets all the protein coding regions in each patient's genome. These datasets will be referred to as 1000g and AtLAs, respectively.

### Linear-scale distributed import

To test the performance of importing genomic variant data (VCF files) into GenomicsDB, we carried out the ETL process for an increasing number of samples for both the 1000g and AtLAs datasets. The ETL process wrote 1000 GenomicsDB partitions to Amazon S3 with the following GenomicsDB loader configurations: 1) column based partitioning, 2) disabled synced writes, 3) 20 parallel VCF files, 4) 1000 cells per tile, 5) compress genomicdb array, 6) 1048576 segment size, 7) ping pong buffering, 8) treat deletions as intervals, 9) size per column partition 43581440, 10) discard missing genotypes, and 11) offload VCF output processing for both datasets. END and GT fields were loaded from AtLAs VCFs. Whole exome sequencing contains additional attributes that are not generated in microarray genotype assays. END and GT fields as well as DP, GQ, AD, and PL fields were loaded for 1000g. These additional fields are found in whole exome sequencing datasets, but not in microarray genotype assays. This process was repeated on four different AWS instance types in order to evaluate the effects of resources on import time. We used two general-purpose instances m4.2xlarge (2 cores, 8G memory) and m4.4xlarge (16 cores, 64G memory) and two memory-optimized instances r4.2xlarge (8 cores, 61G memory) and r4.4xlarge (16 cores, 122G memory) [32].

We found linear-scale ETL times for 1000g (Fig 6) and better than linear-scale ETL times for AtLAs (Fig 7) as the number of samples increase. The import times for AtLAs suggest logscale potential, although more samples would be required to determine this with confidence. The import times for both 1000g and AtLAs suggest that m4.4xlarge and r4.4xlarge instances result in shorter import times with increasing sample size in comparison to the m4.2xlarge and r4.2xlarge instances. This result is expected since the m4.4xlarge and r4.4xlarge have more processing power than the m4.2xlarge and r4.2xlarge instances. A less expected result was that there were no significant advantages to using the memory-optimized nodes for ETL. This result suggests that the lower cost, general-purpose nodes provide sufficient resources to perform the ETL process efficiently with the provided configurations.

### Linear-scale database partition size

To evaluate how total GenomicsDB array size grows as the number of samples in the database increases, we measured the total partition size of the databases resulting from the ETL processes described above (sum of 1000 partitions). Storage size increased linearly with the number of samples for both 1000g and AtLAs, with the storage size for 1000g increasing at a faster rate (Fig 8). This observation is expected since the 1000g dataset is importing more fields from the VCF files and has more coverage of the genome, as compared to the AtLAs dataset.

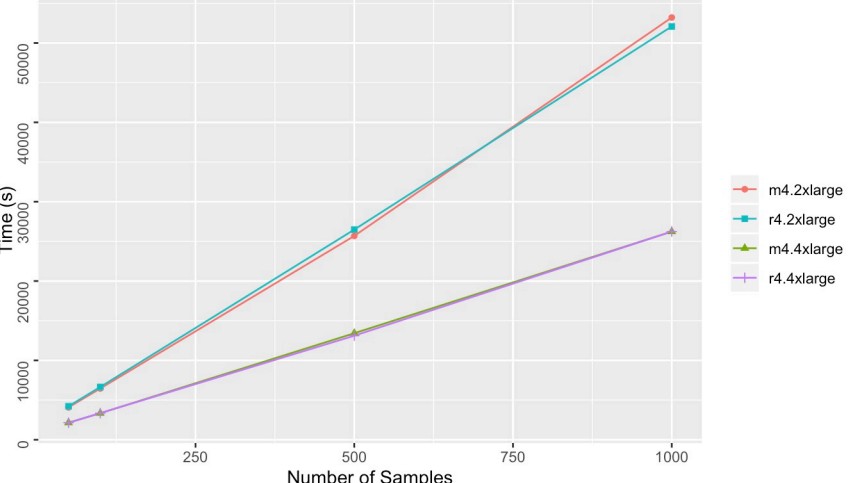

**Fig 6. Whole exome import times.** Time to write 1000 GenomicsDB partitions to Amazon S3 from 1000 genomes VCF file inputs. Measurements were taken for an increasing sample size, and for four types of AWS instance types (m4.2xlarge, r4.2xlarge, m4.4xlarge, and r4.rxlarge).

## Linear-scale genotype query time

To test the performance of Spark-based GenomicsDB queries, we queried variable-sized regions across the genome of both datasets using the GDBSpark API. Regions include a large chromosome (chromosome 1—add # bp), a medium sized chromosome (chromosome 10—add # bp), a small chromosome (chromosome 22—add #bp), a set of 3 genes on 3 different chromosomes, a single gene, and two point mutations. The smaller regions (mutations and genes) were chosen to mimic query functionality that is likely to come from a targeted exploration (ie. within i2b2). The larger regions (chromosomes) were chose to reflect variables sizes of genomic regions that would come from more general exploration, such as genome wide

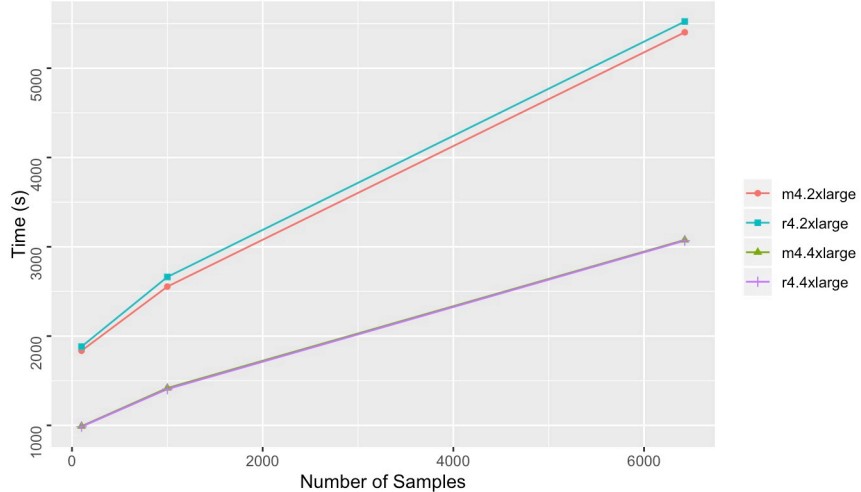

**Fig 7. Microarray import times.** Time to write 1000 GenomicsDB partitions to Amazon S3 from AtLAs VCF file inputs. Measurements were taken for an increasing sample size, and for four types of AWS instance types (m4.2xlarge, r4.2xlarge, m4.4xlarge, and r4.rxlarge).

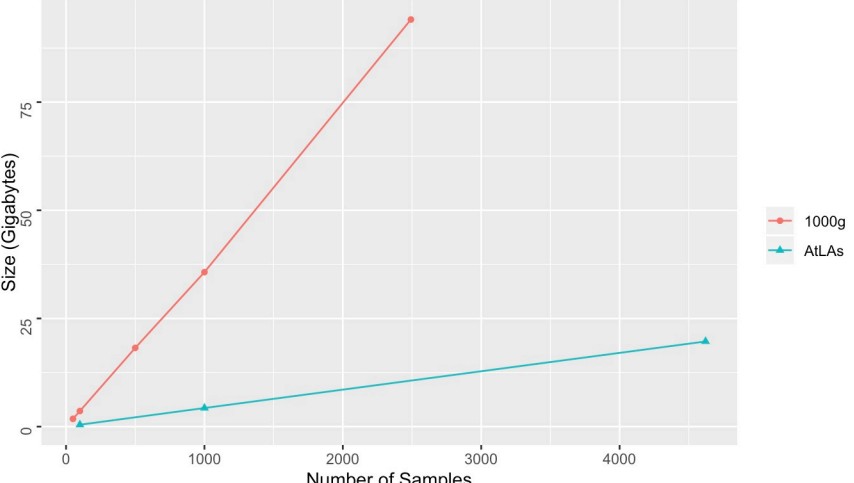

**Fig 8. Samples by partition size.** Total partition size (sum) as total number of samples in the database increases. Results are shown for both 1000 genomes and AtLAs datasets.

association studies. Similar to the above experiments, query times were measured for an increasing number of samples in the database. Each query requested data for all samples in the respective GenomicsDB instance. Queries were performed five times and the average was reported. A query within the GDBSpark API is the time it takes for data to be retrieved from GenomicsDB, imported into an RDD in VariantContext format, perform a user-specified analysis, and return the result. The reported times are the total time to query GenomicsDB, load the data into an RDD, count the number of variants, and return both the data and the count.

For 1000g dataset, smaller regions exhibit worst-case linear scaling and chromosomes display near-linear scaling (Fig 9). This result is consistent with previously reported results in the GenomicsDB white paper [7]. When more data is read into memory, the number of cache

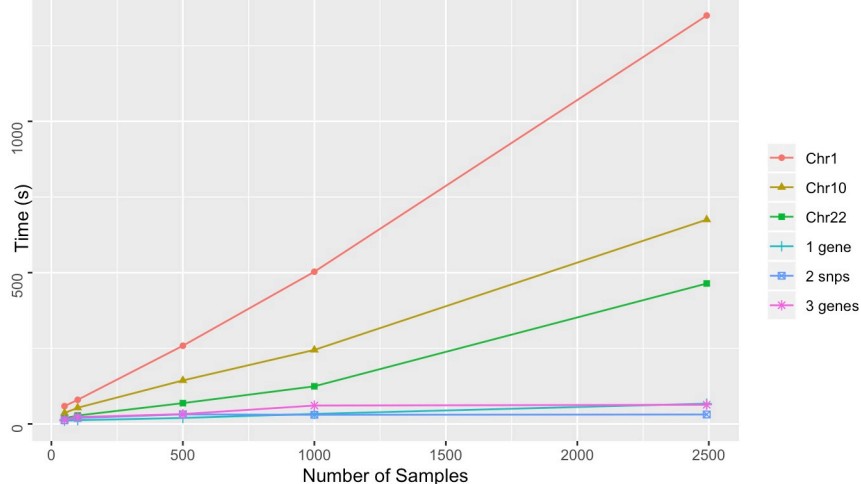

**Fig 9. Whole exome query time.** Time to query varying size GenomicsDB instances for the 1000g dataset. Queries requested all samples in the database and were measured for varying size genomic regions.

misses leads to in-memory bandwidth saturation and causes an increase in read times. The AtLAs queries appear to be reporting near log-scale query times with smaller regions, small chromosomes, and even medium size chromosomes (Fig 10). The larger chromosome for the AtLAs dataset is close to linear, but still exhibits a trend that is suggestive of the memory-saturation behavior of GenomicsDB.

## Linear-scale genotype-phenotype query time

To test the performance of clinical and genomic data integration within the framework, we measured the time to import clinical datasets for an increasing number of patients and the time to perform an aggregate query. Clinical datasets were generated from the clinical information associated to the AtLAs variant data associated to over 12,000 patients in the UCLA i2b2 CRC cell. These datasets were created by selecting patients who consented to sharing genomic information and were classified in one of the following cohorts: i) C50 malignant neoplasms of breast (754 samples), ii) consenting patients with ICD-10 C00-D49 neoplasms (1989 samples), iii) all consenting male patients (5105 samples), iv) patients in the diagnosis class of J00-J99 diseases of the respiratory system who have been prescribed some throat preparation medication (8942 samples), v) K00-K95 diseases of the digestive system (9839 samples), and patients who have recorded information for "Various" medications (10405 samples). Requests were sent from the web browser of i2b2 for a subset of ICD10 ontological terms and the response was saved to an XML file. The time to import this XML file into Spark and count the number of patients in the set, collectively, was recorded five times for each query and the average was reported. Worst-case linear time was observed for an increasing number of patients (Fig 11).

The time required to associate this clinical set to the genomic data, query GenomicsDB, collate the integrated set into the distributed environment, and perform a simple aggregate (count) calculation, collectively, was also recorded. We use a subset of the genomic regions queried above for these queries, and report these regions based on the number of variants returned. Each query was performed five times and the average was reported. Again, we observe worst-case linear query response time for the aggregation of the clinical and genomic

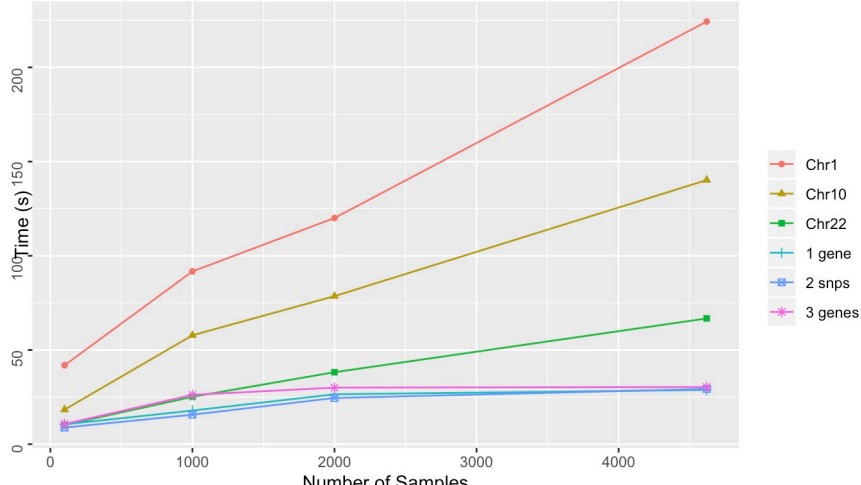

**Fig 10. Microarray query time.** Time to query varying size of GenomicsDB instances for the AtLAs dataset. Queries requested all samples in the database and were measured for varying size genomic regions.

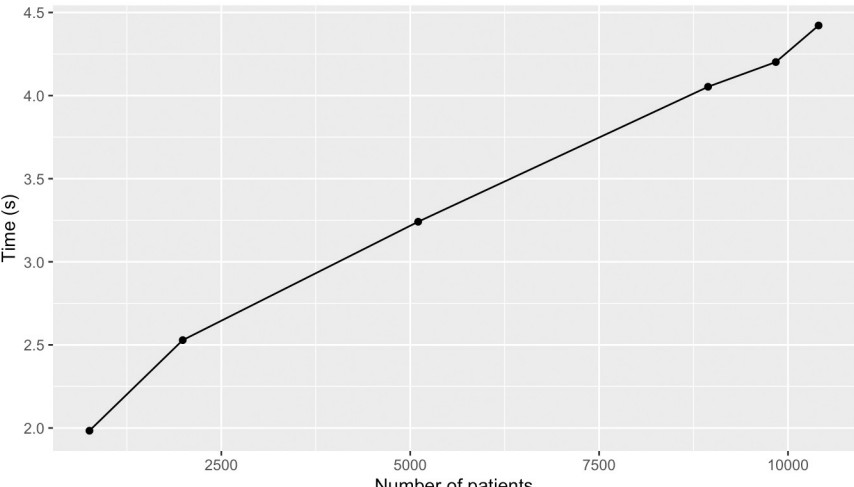

**Fig 11. Patients by clinical dataset import time.** Time required to import i2b2 derived clinical datasets into distributed environment for an increasing number of patients.

data for an increasing number of regions queried (Fig 12). The number of patients in the query had less of an effect on query time, but still exhibited linear scaling (Fig 13) with respect to the number of patients queried. This result suggests that response time is affected only minimally by the size of the patient set query.

These queries were performed for a subset of samples in a constant size GenomicsDB instance. This is in contrast to the previously presented experiment, which queried all the samples in a given instance. It is important to note that during the short time between performing these two experiments, the number of samples in the system had nearly doubled in size. These results show that the worst-case linear performance is preserved even for clinical-genomic

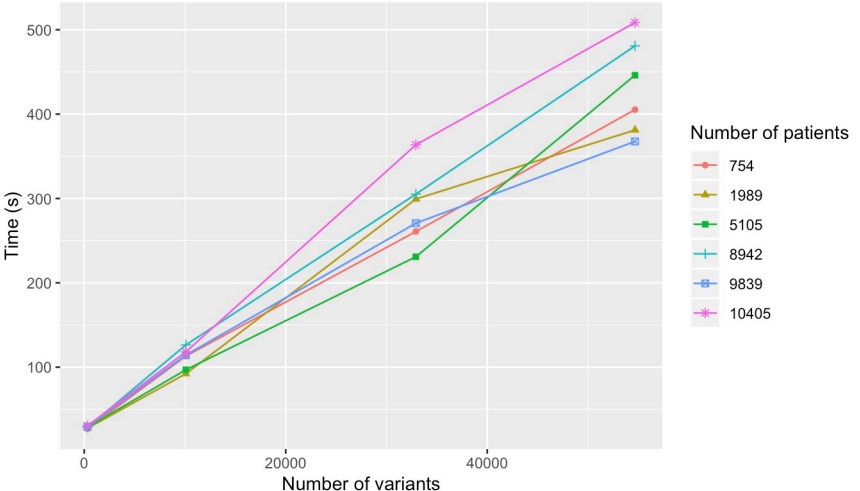

**Fig 12. Variants by genotype-phenotype query time.** Time to aggregate the clinical information from the i2b2-derived file, associate to the samples in the genomic database, load genomic variant data into distributed environment, and count the number of variants. Results are shown for an increasing number of variants queried for varying sets of patient sizes.

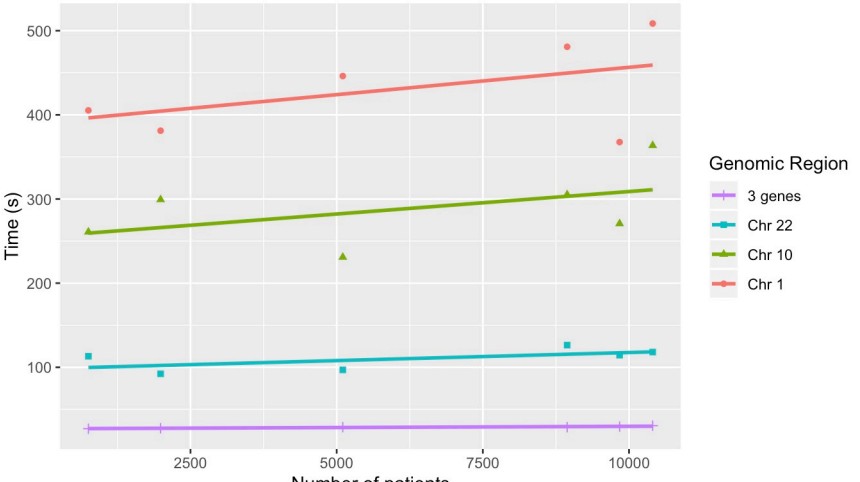

**Fig 13. Patients by genotype-phenotype query time.** Clinical and genomic aggregation time as the number of patients increases, displayed for varying sets of genomic regions.

aggregate queries for over 10,000 patients. The results also show that the number of variants queried will have more of an effect on response time than the number of patients referenced in the query. This means that more patients could be included in an analysis without having a significant impact on query response time.

## Discussion

Throughout this paper we present a distributed analytics framework for performing precision medicine queries. We use a real-world use case from UCLA to drive the development of the framework and confirm scalability and efficiency of distributed loading, storage, access, and aggregate queries for an increasing number of samples. The results have shown linear query times from distributed GenomicsDB instance with an HDFS-compliant file system. These results were made possible by extending GenomicsDB to support reading and writing to a distributed file system, which has resulted in a more Spark friendly version of GenomicsDB. Despite network latencies associated with HDFS and Spark, the results show that the distributive power maintains, and often improves upon, previously reported GenomicsDB results that use a native file system for both microarray and whole exome sequencing data. If processing samples are to be expected to be thousands weekly, then scalability with increasing patient data added to the system is of great importance for import, storage, and query times. Further, the ODA platform maintains scalability in the downstream analytics environment to scale at all points of the analysis pipeline.

Our extensions to GenomicsDB have several maintenance and cost related advantages in addition to those noted in the above results section, which we have observed from utilizing AWS. First, the ability to read and write from a distributed file system means the GenomicsDB arrays can be stored and queried from cost-effective Amazon S3 buckets rather than requiring each slave node to have an additional EBS mount to store the data locally. Second, our extensions allow for an increase in the number of RDD partitions per worker without adding new slave nodes which requires a smaller and more cost effective EMR instance than the setup that allowed for only one RDD partition per worker. By persisting GenomicsDB arrays on Amazon S3, we are able to maintain data integrity and fault tolerance in the event of a failed worker

node. These extensions to GenomicsDB allow for better, more on-demand resource balancing. Without the extensions described in this paper, the management of GenomicsDB for use on a Spark cluster would be extremely difficult to maintain.

We have also shown performance advantages to using GenomicsDB within a distributed processing environment. There appears to be a strong correlation between import and query times and the amount of data involved in the process. The targeted-sequencing dataset has shown log-scale potential for import, and near log-scale potential for query execution. For import and query operations involving a small amount of data, the setup costs will not be fully amortized and the static-partitioning scheme used to construct GenomicsDB arrays could be introducing tasks that are much larger than others leading to bottlenecks. All query regions, other than the largest chromosome, for the target-sequencing data exhibit log-scale behavior, which is likely due to all the data fitting into Spark and/or GenomicsDB cache. The linear performance for the larger dataset indicates some overhead involved in not fitting all the data into memory. The linear import and query times exhibited for whole exome sequencing data suggests that even the smaller queries have enough data to not be burdened by setup time, and thus take full advantage of the distributed processing environment. Given the largest query region shows linear behavior for both the microarray data and the, much heavier, whole exome data, we would expect this trend to continue for, even heavier, whole genome data. Though, further exploration will need to confirm this definitively.

Future plans for GenomicsDB include extending the database to be more Spark-friendly in terms of file system support, database distribution, and partitioning schemes. GenomicsDB arrays are statically partitioned during the import process by splitting the genome into a user-specified amount of chunks. The distribution of variants is not even across the genome space, meaning that some arrays contain a lot more variant data than others. This can lead to load imbalance across both data ingesting and querying. Proper load balancing of a Spark application can lead to better execution times of aggregate queries. More advanced ways to distribute the variant data into GenomicsDB arrays should be considered, such as sampling the variant data before loading to understand mutation burden across the genome. Ideally, we would like to create an array distribution that optimizes load balancing and provides support for loading a variable amount of RDD partitions—potentially dependent on the resources available to the application. Other areas of exploration should include how additional data types impact the ability to scale GenomicsDB, such as whole genome sequencing and other genomic data modalities.

Though GenomicsDB and the interaction with Spark was the main focus of this work, the API, mapping database, and user applications are presented as an example of how GenomicsDB can be used to scale out precision medicine platforms at existing hospitals. For instance, we introduced the i2b2 VariantView plugin that enables clinicians to perform on-demand, clinical-genomic precision medicine queries. The plugin uses the ODA Framework to submit a distributed, aggregate query inside a HIPAA-compliant, virtual private network at UCLA David Geffen School of Medicine. The application loads clinical data from i2b2, queries distributed GenomicsDB partitions that reside on Amazon S3, performs a specified analysis report within the distributed environment, and write the results back for visualization in the i2b2 browser interface. Previous efforts to support genomic-based queries in i2b2 have reported linear scaling only up to 500 samples [12]. Our results have exhibited worst-case linear scaling with over 2400 samples from the 1000 genomes whole exome sequencing data and over 10,000 patients with targeted-sequencing data. Further, to the best of our knowledge this is the first effort to extend i2b2 to run Spark-based, distributed queries and processing from within the i2b2 interface.

The prototype system described in this paper is currently designed to support generic analysis applications from the i2b2 front end. However, the applications available to front end users are limited to simple proof of use allele and clinical concept distribution charts. The validation and usability of these applications will only be proven with use of the system by both clinicians and researchers. We anticipate integrating more complex analytical tools into the system that will improve cohort selection through means of more advance statistics and machine learning techniques. Many of the claims made about scalability should be continuously validated as more samples and more data is added to the system for both microarray and more large-scale, next generation sequencing data.

## Conclusion

The power of precision medicine is dependent on the ability to combine data across multiple types and sources to enable quick and scalable joint analyses that support cohort selection and analysis. We have presented an efficient and scalable means for genomic-based cohort exploration and analysis using an optimized genomics database. We show that data can reside in optimal data stores, while still supporting scalable, distributed analytics. Our extensions to GenomicsDB provide support with a distributed file system to provide ease of interaction from a distributed compute environment as well as cost advantages for hosting-related, hardware requirements. The ODA framework can be integrated into existing code for advanced usage, or can be used to extend a HIPPA compliant clinical interface (EHR or cohort system) to execute distributed aggregate queries. Scalable and efficient data processing platforms and databases, such as the framework described in this paper, will be necessary to drive precision medicine forward as data grows and analysis becomes more complex.

## Supporting information

**S1 Repository. GenomicsDB.** https://github.com/GenomicsDB/GenomicsDB.
(DOCX)

**S2 Repository. VariantView plugin.** https://github.com/OmicsDataAutomation/i2b2-oda-framework.
(DOCX)

**S3 Repository. Raw runtimes.** 10.6084/m9.figshare.11796126.
(DOCX)

## Acknowledgments

Thanks to the UCLA Institute for Precision Health who provided continued oversight and support for this project, as well as the clinicians and researchers at the David Geffen School of Medicine at UCLA (DGSOM) who provided input and feedback throughout the duration of this work. Additional thanks to the Digital technology team at DGSOM and the UCLA CTSA I2B2 support team who helped in the deployment and testing of the framework at UCLA.

## Author Contributions

**Conceptualization:** Jaclyn M. Smith, Melvin Lathara, Ganapati Srinivasa, Christopher T. Denny.

**Data curation:** Jaclyn M. Smith, Christopher T. Denny.

**Formal analysis:** Jaclyn M. Smith.

**Funding acquisition:** Ganapati Srinivasa, Christopher T. Denny.

**Investigation:** Jaclyn M. Smith.

**Methodology:** Jaclyn M. Smith, Melvin Lathara.

**Project administration:** Jaclyn M. Smith, Melvin Lathara, Ganapati Srinivasa, Christopher T. Denny.

**Resources:** Jaclyn M. Smith.

**Software:** Jaclyn M. Smith, Melvin Lathara, Hollis Wright, Brian Hill, Nalini Ganapati.

**Supervision:** Jaclyn M. Smith, Ganapati Srinivasa, Christopher T. Denny.

**Validation:** Jaclyn M. Smith, Melvin Lathara, Hollis Wright, Nalini Ganapati, Christopher T. Denny.

**Visualization:** Jaclyn M. Smith.

**Writing – original draft:** Jaclyn M. Smith, Brian Hill.

**Writing – review & editing:** Jaclyn M. Smith, Melvin Lathara, Hollis Wright, Nalini Ganapati.

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
