## [Decision Letter · Decision Letter 0]

11 Dec 2019

PONE-D-19-32466

Advancing clinical cohort selection with genomics analysis on a distributed platform

PLOS ONE

Dear Ms Smith,

Thank you for submitting your manuscript to PLOS ONE. Both reviewers found your work represent an important technical advance, but they also raised some concerns. Particularly, they suggested that example workflows and demos should be included and/or made public. Therefore, we invite you to submit a revised version of the manuscript that addresses the points raised during the review process. 

We would appreciate receiving your revised manuscript by Jan 25 2020 11:59PM. To enhance the reproducibility of your results, we recommend that if applicable you deposit your laboratory protocols in protocols.io, where a protocol can be assigned its own identifier (DOI) such that it can be cited independently in the future. For instructions see: http://journals.plos.org/plosone/s/submission-guidelines#loc-laboratory-protocols

We look forward to receiving your revised manuscript.

Kind regards,

Jianjiong Gao

Academic Editor

PLOS ONE

Journal Requirements:

1. Thank you for including your competing interests statement; "J Smith, M Lathara, H Wright, B Hill, N Ganapati, and G Srinivasa are sponsored by Omics Data Automation, Inc. Competing interests have been fully disclosed and arranged through contract and licensing agreements with UCLA David Geffen School of Medicine. The authors declare that they have no other competing interests."

We note that you received funding from a commercial source:Omics Data Automation, Inc.

Reviewers' comments:

Reviewer's Responses to Questions

**Comments to the Author**

1. Is the manuscript technically sound, and do the data support the conclusions?

Reviewer #1: Yes

Reviewer #2: Yes

2. Has the statistical analysis been performed appropriately and rigorously? 

Reviewer #1: N/A

Reviewer #2: Yes

3. Have the authors made all data underlying the findings in their manuscript fully available?

Reviewer #1: Yes

Reviewer #2: Yes

4. Is the manuscript presented in an intelligible fashion and written in standard English?

Reviewer #1: Yes

Reviewer #2: Yes

5. Review Comments to the Author

Reviewer #1: This paper provides a nice technical write up of extending GenomicsDB to handle distributed file systems and benchmarking on a real set of genomic data associated to clinical data. Notably including cloud-native work to use S3 which provides resiliency and potential for the underlying data to be reused by other applications. Profiling was performed for both the ingest of the data, which is often overlooked, and the query times. This gave a good sense of how the system would perform in reality with microarray and whole exome data. Finally, an example of a potential application tied with clinical data querying in i2b2 is described.

Fairly minor adjustments that may benefit the paper:

1) The authors note that the microarray data has "log-scale potential" while the exome data is linear. Further insight into why this might be would be useful for the audience, especially as increasingly whole genome data is available and would be interesting to understand if the expectation of linear time would still hold?

2) It would be nice to include the specific queries run as well as the descriptions. The main use case here is for regions of genomes - were other forms of queries considered?

3) While the extension itself is in Github as part of GenomicsDB, the GDBSpark API, i2b2 and Zeppelin demo are all available "upon request". I would highly encourage the authors to make this available on Github in some form as well, even if the licensing only covers academic use.

Reviewer #2: This paper by Smith et. al. represents an important advance in the interactive analytic methods for phenotype and genotype to be mutually interacted upon and enable the basis of nearly real time query capability. The unification of i2b2 and GenomeDB means that a powerful phenotyping database and a powerful genotyping database can operate in concert to enable researchers and clinicians to actively explore the observational data flowing from clinical care.

The architecture of the application takes advantage of the strengths of both the i2b2 and the GenomeDB platforms by connecting the two using an i2b2 plug-in. This always the solution to be maintainable as the backward compatibility of the i2b2 platform is guaranteed for its web services that interconnect the plug-ins.

Overall the paper does an excellent job conveying the technical features of how GenomeDB provides a scalable and interoperable platform for genome analysis with i2b2, however it would make the paper more readable if the paper included some example workflows for specific phenotype/genotype query lines. By this is meant that example queries and analyses could be presented to more clearly show how the tools could be used in the course of discovery of new genomic variant associations with phenotypes, and analysis of GWAS and PheWAS experiments. Especially useful would be some screen shots along the way to show the evolution of the analysis to give the reader the sense of how the queries and output would look to the researcher using the tools.

6. PLOS authors have the option to publish the peer review history of their article (what does this mean?). If published, this will include your full peer review and any attached files.

Reviewer #1: No

Reviewer #2: No

---

## [Author Response · Author response to Decision Letter 0]

3 Feb 2020

All responses to the reviewer can be found in the Response to Reviewers letter. The responses are provided below:

Thank you for facilitating the prompt review of our manuscript entitled “Advancing Clinical Cohort Selection with Genomics Analysis on a Distributed Platform” by Smith et al, that was submitted for consideration in the digital heath call for papers. We were very glad that you and the reviewers felt our work represented an important technical advance. Using reviewer feedback as a guide, we have made further improvements to the manuscript that we have itemized as follows: 

Reviewer 1. Overall comments were very positive. Reviewer 1 did include a short list of suggested “minor adjustments that may benefit the paper”. We respond to them in order as follows:

1) Suggested clarification regarding the log-scale potential of microarray data and how these results could expand into whole genome sequencing datasets. 

This is an important point, and we have extended the discussion section (page 24) to reflect this. In summary, the results suggest strong correlation between import and query time and the amount of data involved. Smaller data sizes will suffer more from start up costs without much gain, whereas the larger datasets are able to take more advantage of the distributed processing environment. Given the microarray data starts to show linear behavior for the largest query region, and that the much heavier, whole exome data continues to show linear behavior for this query, suggests that this trend will continue for heavier, whole genome data.

2) Suggested including specific queries run with the I2B2 plugin module.

We have extended the introduction (pages 5,6) with I2B2 workflow examples that can be used as motivating examples. In addition, the experiments section now describes further clinical filters that were explored (page 21), and provides an explanation for the types of genomic regions considered (page 19).

3) Making code for the I2B2 plugin and the GDBSparkAPI available for academic use was highly encouraged.

We very much agree with this sentiment. Facilitating implementation and use of these tools in a generalizable way in academic settings, has been a goal of this project from its inception. With this in mind, we have made a version of the i2b2 plugin available under the MIT license, and the GDBSparkAPI and gdb-mapping database available for academic use. These components can be found at https://github.com/OmicsDataAutomation/i2b2-oda-framework. All GenomicsDB components are available at: https://github.com/GenomicsDB/GenomicsDB for free academic use under the MIT license. I2B2 patient querying related components are available at https://www.I2B2.org/software/index.html.

Reviewer 2. Overall comments were again very positive. This reviewer also requested example queries demonstrating the potential utility of these tools to researchers. We address this in item 2) above. 

Once again, thank you for the prompt and thorough reviews. We very much hope that you now find our revised manuscript to be suitable for publication in PLOS ONE. We look forward to hearing from you at your earliest convenience.

---

## [Decision Letter · Decision Letter 1]

2 Apr 2020

Advancing clinical cohort selection with genomics analysis on a distributed platform

PONE-D-19-32466R1

Dear Dr. Smith,

We are pleased to inform you that your manuscript has been judged scientifically suitable for publication and will be formally accepted for publication once it complies with all outstanding technical requirements.

With kind regards,

Jianjiong Gao

Academic Editor

PLOS ONE

Additional Editor Comments (optional):

Reviewers' comments:

Reviewer's Responses to Questions

**Comments to the Author**

1. If the authors have adequately addressed your comments raised in a previous round of review and you feel that this manuscript is now acceptable for publication, you may indicate that here to bypass the “Comments to the Author” section, enter your conflict of interest statement in the “Confidential to Editor” section, and submit your "Accept" recommendation.

Reviewer #1: All comments have been addressed

2. Is the manuscript technically sound, and do the data support the conclusions?

Reviewer #1: Yes

3. Has the statistical analysis been performed appropriately and rigorously? 

Reviewer #1: N/A

4. Have the authors made all data underlying the findings in their manuscript fully available?

Reviewer #1: Yes

5. Is the manuscript presented in an intelligible fashion and written in standard English?

Reviewer #1: Yes

6. Review Comments to the Author

Reviewer #1: (No Response)

7. PLOS authors have the option to publish the peer review history of their article (what does this mean?). If published, this will include your full peer review and any attached files.

Reviewer #1: No

---

## [Editor Report · Acceptance letter]

7 Apr 2020

PONE-D-19-32466R1 

Advancing clinical cohort selection with genomics analysis on a distributed platform 

Dear Dr. Smith:

I am pleased to inform you that your manuscript has been deemed suitable for publication in PLOS ONE. Congratulations! Your manuscript is now with our production department. 

With kind regards,

on behalf of

Dr. Jianjiong Gao 

Academic Editor

PLOS ONE